# Modification and Application of Bamboo-Based Materials: A Review—Part I: Modification Methods and Mechanisms

**Zhiyu Zheng** [1], **Nina Yan** [1], **Zhichao Lou** [2,*], **Xizhi Jiang** [1], **Xiaomei Zhang** [1], **Shan Chen** [1], **Rui Xu** [1], **Chun Liu** [1] **and Lei Xu** [1,*]

1. Jiangsu Engineering Technology Research Center of Biomass Composites and Addictive Manufacturing, Institute of Agricultural Facilities and Equipment, Jiangsu Academy of Agricultural Sciences, Nanjing 210014, China; zhiyu7875@gmail.com (Z.Z.); yannina@jaas.ac.cn (N.Y.); jiangxizhi@jaas.ac.cn (X.J.); zhangxiaomei@jaas.ac.cn (X.Z.); chenshan@jaas.ac.cn (S.C.); 20210080@jaas.ac.cn (R.X.); 20180075@jaas.ac.cn (C.L.)
2. Jiangsu Co-Innovation Center of Efficient Processing and Utilization of Forest Resources, Nanjing Forestry University, Nanjing 210037, China
* Correspondence: zc-lou2015@njfu.edu.cn (Z.L.); xulei@jaas.ac.cn (L.X.)

**Abstract:** In light of continual societal advancement and escalating energy consumption, the pursuit of green, low-carbon, and environmentally friendly technologies has become pivotal. Bamboo, renowned for its diverse advantages encompassing swift growth, ecological compatibility, robust regenerative properties, commendable mechanical characteristics, heightened hardness, and abundant availability, has discovered applications across various domains, including furniture and construction. Nevertheless, natural bamboo materials are plagued by inherent limitations, prominently featuring suboptimal hydrophobicity and vulnerability to fracture, thereby constraining their broad-scale application. Thus, the paramount concern is to enhance the performance of bamboo materials through modification. However, prevailing reviews of bamboo modification predominantly concentrate on physical or chemical approaches, resulting in a conspicuous absence of a comprehensive overview of bamboo modification techniques. This review explores an array of bamboo treatment modalities and delivers a valuable assessment of bamboo modification, offering significant guidance for forthcoming bamboo enhancement and utilization endeavors.

**Keywords:** bamboo; sustainability; modification

## 1. Introduction

With the intensification of global climate change and environmental pollution, embracing the path of green, low-carbon, and sustainable development is the inevitable choice for humanity [1–3]. As countries around the world impose increasingly stringent requirements on carbon emissions, bamboo resources, often referred to as "carbon-negative" materials, have become notably significant [4–6]. The process of bamboo growth itself is a carbon reduction process, as it can absorb carbon dioxide from the air, convert it into carbohydrates through photosynthesis, and release oxygen. Enhancing the utilization value of renewable bamboo forest resources, substituting traditional metals and petrochemical-based functional materials is one of the most feasible methods to reduce carbon emissions, increase carbon sequestration, and maintain atmospheric carbon balance [7].

Bamboo, an intriguing and adaptable natural resource, is distinguished by its swift growth, distinctive structure, and composition. In the realm of production, bamboo showcases remarkable growth rates, with certain species capable of achieving several inches of growth daily. Consequently, bamboo lends itself to sustainable harvesting, and its production cycle notably outpaces that of many other materials. This attribute underpins its standing as a readily renewable resource [8]. The structure of bamboo is a key component of its appeal. It comprises hollow segments, or "culms", joined by nodes, providing

flexibility and strength. This segmented structure is not only aesthetically pleasing but also offers exceptional mechanical properties, making it a favored choice in construction and engineering [9–14]. The composition of bamboo includes cellulose, hemicellulose, and lignin, similar to traditional hardwoods [15]. However, the proportion of these components can vary by species. The high cellulose content contributes to its strength, while lignin enhances its durability and resistance to pests and decay. The unique combination of these constituents is part of what makes bamboo a valuable resource for various applications. The remarkable production rate, segmented structure, and versatile composition of bamboo make it a valuable and sustainable resource. Its unique qualities have led to a wide range of applications, from construction and furniture to textiles and eco-friendly products. Bamboo's prominence as a renewable and adaptable material underscores its significance in addressing environmental and industrial challenges.

However, certain inherent characteristics of natural bamboo materials impose limitations on their applications. Firstly, due to the presence of numerous free hydroxyl groups and their porous structure, natural bamboo materials tend to absorb moisture, resulting in poor dimensional stability [16,17]. This property implies that, under humid or wet conditions, bamboo materials are prone to expansion or contraction, which is adverse for various applications such as construction and furniture manufacturing that demand superior dimensional stability. Secondly, natural bamboo materials exhibit relatively low resistance to biological factors, making them susceptible to fungal, bacterial, and pest attacks [18]. This susceptibility can lead to bamboo decay or degradation, ultimately reducing their lifespan. Lastly, the mechanical properties of natural bamboo materials are relatively unstable due to their porous structure. Different parts of bamboo may exhibit varying mechanical properties, and even within the same bamboo pole, a gradual decrease in strength can be observed from the outer to inner sections [19,20]. These drawbacks collectively contribute to a reduced lifespan of natural bamboo materials and limit their range of applications. Therefore, it becomes essential to subject natural bamboo materials to appropriate treatments to meet the diverse requirements of various applications.

Researchers have been actively engaged in innovative bamboo-based material modification studies, imparting entirely new characteristics and performance capabilities to bamboo. This enables bamboo to serve as a versatile alternative to traditional petroleum-dependent derivative materials. This exploration in the field is primarily driven by sustainability considerations, aiming to propel the development of green materials and eco-friendly products. In recent years, novel modification methods (heat treatment, chemical modification, plasma, and microwave modification etc.) have continuously emerged in the research field with the aim of imparting bamboo with multifunctionality and superior performance [21–23]. For instance, Feng et al. subjected 4-year-old Moso bamboo to superheated steam treatment at 160–220 °C for 1.5 h, resulting in improved hydrophobicity and dimensional stability of the bamboo [24]. Huang et al. employed a chemical modification method that combined vinyl acetate acetylation and methyl methacrylate in situ polymerization. The results indicated an enhancement in the dimensional stability, thermal stability, and wettability of the modified bamboo [25]. These innovative methods include enhancing bamboo fiber strength, improving its water resistance, and increasing its stability, thereby expanding the scope of bamboo's applications in areas such as construction, materials science, and environmental engineering. This potential for multifunctionality further solidifies bamboo's role in sustainable development and the realm of green materials.

Up to now, common techniques used to enhance the performance of natural bamboo materials include chemical, physical, physicochemical, and biological treatments [26–28]. However, each of these methods has its own advantages and drawbacks, and researchers must choose the most suitable, cost-effective, and efficient modification approach based on their specific applications. For instance, biological treatments, such as enzymatic treatment, are relatively expensive and not practical for large-scale processing [29]. Consequently, chemical treatments, including alkali and acid treatments, are more commonly used [30]. Nevertheless, chemical treatments can generate harmful byproducts, leading to environ-

mental pollution and potential health hazards [31,32]. In contrast, physical treatment methods are favored for their eco-friendliness, simplicity, and safety but there are limitations [33]. When selecting the most suitable modification technique for bamboo based on specific requirements and application areas, the following criteria and considerations should be primarily taken into account. On the one hand, it is essential to identify the specific performance characteristics needed for the application, such as strength, durability, water resistance, fire resistance, and appearance. On the other hand, evaluate the environmental sustainability of the modification technique. Choose methods with minimal environmental impact that align with sustainable development goals, and also consider the cost-effectiveness of the modification. This article provides an overview of common treatment methods for natural bamboo materials in recent years, summarizing their impacts on the performance of bamboo materials. Additionally, this review serves not only as a valuable resource for individuals and businesses engaged in bamboo material research, but also provides clear guidance for future endeavors in this field.

## 2. Thermal Treatment

Bamboo heat treatment can be categorized based on the heat transfer medium used. It includes steam heat treatment, oil heat treatment, and air or inert gas heat treatment. Among these, steam heat treatment and oil heat treatment have gained significant popularity in both research and industry [18].

### 2.1. Steam Heat Treatment

Steam heat treatment involves using steam as a medium to treat bamboo. Steam serves the dual purpose of softening bamboo and isolating it from oxygen exposure. Depending on the specific characteristics of steam, steam heat treatment can be further categorized into saturated steam treatment and superheated steam treatment. The equipment diagrams of superheated steam treatment and saturated steam treatment are shown in Figure 1. Saturated steam is generated when the evaporation and condensation processes are in dynamic equilibrium during water evaporation. It is characterized by a one-to-one correspondence between temperature and pressure. Saturated steam, due to its remarkable penetrating ability and capacity to lower the glass transition temperature, can soften bamboo at relatively low temperatures while minimizing the reduction in bamboo's mechanical properties. Notably, saturated steam heat treatment is highly efficient and environmentally friendly, making it one of the foremost methods for bamboo modification in the realm of heat treatments [34–36].

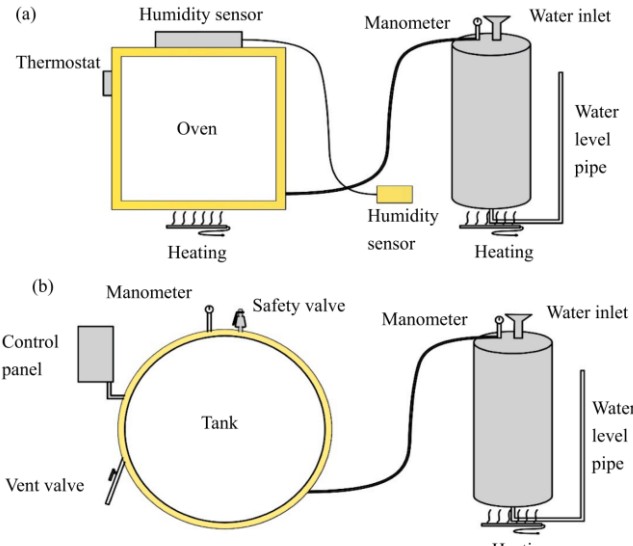

**Figure 1.** (**a**) Superheated steam treatment equipment, (**b**) saturated steam treatment equipment [18].



For example, Yuan et al. aimed to assess the impact of saturated steam treatment on bamboo, exploring the influence of temperature and duration. Bamboo samples were exposed to saturated steam at temperatures of 140 °C, 160 °C, and 180 °C for durations of 4, 6, 8, and 10 min, with an untreated control group. Findings revealed that at temperatures above 160 °C, hemicellulose and cellulose content decreased, while lignin content increased, reducing the equilibrium moisture content and enhancing hygroscopicity. Mechanical properties were significantly affected, with 140 °C treatment increasing the modulus of rupture (MOR) and modulus of elasticity (MOE), but at 180 °C, 10 min treatment resulted in MOR and MOE reductions. These findings have implications for bamboo modification and various applications [34]. Superheated steam treatment was employed to modify Moso bamboo by Feng et al. Under high-temperature hygrothermal treatment, bamboo undergoes a series of positive transformations. It exhibits reduced hygroscopicity, with a 16.5% decrease in volumetric swelling, signifying improved stability in high humidity environments. Additionally, dimensional stability is notably enhanced, as indicated by a three-fold increase in the contact angle at 220 °C, making bamboo suitable for diverse environmental conditions. Furthermore, microstructural changes, including fiber and parenchyma cell separation and pore formation, contribute to improved structural stability. These affirmative experimental outcomes underscore the significant impact of hygrothermal treatment on bamboo's physical properties, rendering it a promising candidate for applications in high-humidity environments [24].

### 2.2. Oil Heat Treatment

Oil heat treatment is a crucial method for bamboo thermal modification, employing vegetable oils like tung oil, linseed oil, palm oil, or mineral oils such as methyl silicone oil as the heat transfer medium [37–41]. Firstly, it achieves efficient heat transfer, ensuring uniform heating throughout the material, thus avoiding issues of temperature irregularities. Additionally, it enables precise temperature control, allowing researchers to finely adjust treatment temperatures as needed for optimal results. This multifunctional and widely applicable method has gained significant attention in the field of bamboo modification.

For example, the impact of methyl silicone oil at different temperatures (140 °C to 200 °C) and various durations (2 h to 6 h) on the chemical composition, physical-mechanical properties, surface wettability, corrosion resistance, and mold resistance of bamboo was systematically investigated by Hao et al. Under the conditions of 2 h of oil heat treatment at 160 °C, the bamboo exhibited the highest parallel grain compressive strength, reaching up to 109.52 MPa, which is 18.63% higher than that of untreated samples. The reason for this performance improvement may be due to the increased crosslinking of lignin polymer during heat treatment. When held at 180 °C for 2 h, the bamboo demonstrated its highest flexural strength and modulus of elasticity values, reaching 142.42 MPa and 12,373.00 MPa, respectively, which was caused by the decreased moisture content at the fiber saturation point and the improved dimensional stability. Finally, as the heat treatment temperature and duration increased, the mechanical properties of bamboo were reduced because the content of cellulose and hemicellulose gradually decreased owing to its poor thermal stability, while the material's resistance to corrosion and mold resistance significantly improved due to the blocked internal channels for nutrient exchange, decreased surface wettability, and the degradation of polysaccharides and starch [40]. Similarly, the influence of thermal oil treatment on the physical-mechanical properties of three Philippine bamboo species was investigated by Manalo et al. The findings revealed that thermal oil treatment led to improvements in water absorption and thickness expansion properties for all three Philippine bamboo species [37]. These studies provide comprehensive process parameters and microscopic mechanisms for bamboo oil heat treatment performance, offering practical guidance for production.

### 2.3. Air or Inert Gas Heat Treatment

Apart from steam and oil, air or inert gases, like nitrogen, can be utilized as heat transfer media in bamboo thermal treatment [26,42,43]. Zhang et al. conducted four cycles of thermal treatment on bamboo in the air, involving seven temperature levels (100 °C to 220 °C) and durations ranging from 1 to 4 h. The results revealed that bamboo experienced an increase in mass loss with rising temperature and extended treatment time, with a maximum reduction of 29.0%. Interestingly, at temperatures below 200 °C, the elastic modulus (MOE) of the samples was minimally affected, and in some cases, it even showed a slight increase compared to the control samples [44]. In contrast, Nguyen et al. conducted thermal treatment of bamboo in a nitrogen atmosphere to enhance its durability and dimensional stability [45]. The variations in surface color of bamboo culms specimens treated in different media are shown in Figure 2. The treatment in the air showed a significant dark color at 170 °C, while the sample treated with nitrogen did not show this change until 190 °C. This is because nitrogen is an inhibiting gas, and the pyrolysis ratio is very low, so there is little change in color. When the temperature reaches 210 °C, all the bamboo strips showed a deep red color. At the same time, the color is easy to change under oil treatment, which is caused by the filling of oil molecules on the surface of the bamboo. These findings suggest that bamboo thermal treatment holds significant potential for enhancing bamboo quality.

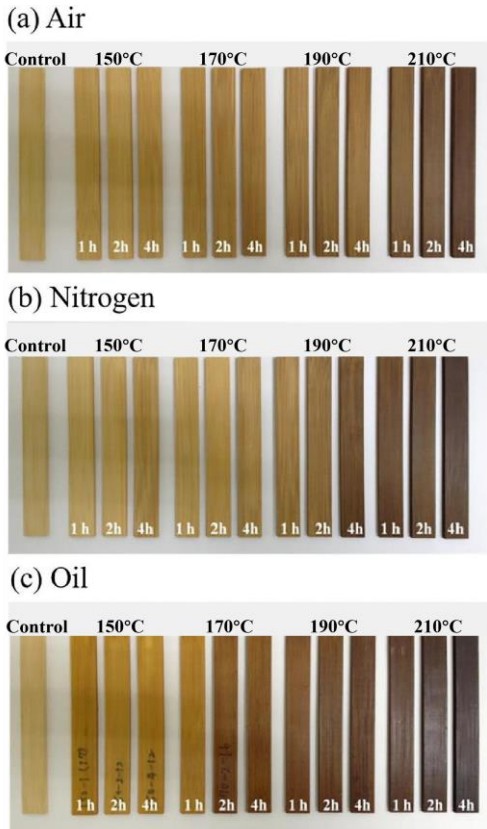

**Figure 2.** Variations in surface color of bamboo culm specimens treated in different media [42].

## 3. Chemical Modification

Chemical modification, a sophisticated approach, entails the utilization of chemical reactions to intentionally transform the chemical composition of bamboo [46–48]. Techniques like alkali and acid treatments, acetylation and in-situ polymerization are employed in this process, enabling precise control over the material's characteristics. These modification methods can significantly boost bamboo strength, making it more robust and durable. Moreover, they enhance its resistance to water, which is particularly crucial for applications

where moisture resistance is vital. Additionally, chemical modification increases bamboo resistance to chemical corrosion, making it suitable for harsh environments. Beyond these mechanical and chemical improvements, chemical modification also holds the potential to modify the surface properties of bamboo, rendering it more compatible with various materials and broadening its scope of application possibilities. This versatility and adaptability of chemically modified bamboo make it a promising contender in the realm of advanced materials.

For example, Chen et al. investigated the effects of alkali treatment on the microstructure, chemical composition, and thermal properties of bamboo parenchyma cells and fibers. The results showed that alkali treatment could partially remove lignin from the parenchyma cells but had little effect on the hemicellulose content, while both lignin and hemicellulose were removed from the fibers [30]. Similarly, their team also examined the effects of alkali treatment on the properties of individual bamboo fibers. Studies showed that alkali treatment with low NaOH concentration (lower than 10%) improved the wettability and thermal stability of bamboo fiber [49]. Wang et al. employed furfuryl alcohol (FA) as a modifier and maleic anhydride (MA) as a catalyst in their study. They prepared various bamboo samples with different levels of FA addition (10 wt%, 20 wt%, and 30 wt%). Detailed investigations were conducted on the physical-mechanical properties of the materials, including weight percent gain (WPG), water uptake (WU), thickness swelling (TS), modulus of rupture (MOR), and modulus of elasticity (MOE). Furthermore, the resistance to decay of both the original bamboo and furfurylated-bamboo samples was studied. The results revealed that FA resin was incorporated into the bamboo and successfully polymerized within the bamboo cell walls. When the FA content reached 30 wt%, significant improvements were observed in all physical properties. Furfurylated bamboo exhibited enhanced thermal stability and decay resistance compared to the original bamboo. Notably, after treatment with 30% FA, furfurylated bamboo experienced only a 5.3% mass loss, reaching a high level of decay resistance [50].

In order to enhance the mildew resistance of bamboo, Chen et al. modified citral and applied it to bamboo processing [51]. The results showed that the resistance rate of modified bamboo to common mold could reach 100%. Similarly, Fan et al. introduced cinnamaldehyde into bamboo by in situ Mannich reaction, which improved the surface hydrophobicity of bamboo and effectively resisted the erosion of mold [52]. Sun et al. employed N-methylol acrylamide (NMA), a cross-linking monomer with dual functional groups, for the modification of bamboo to enhance its resistance to mold. When the monomer concentration reached 6% or higher, the surface of NMA-treated bamboo exhibited no fungal growth, as shown in Figure 3. The main reasons for the improvement of the resistance to mold were the coating effect of PNMA on starch and the antibacterial property of PNMA. Notably, the thermal stability of bamboo remained relatively unchanged before and after modification. The results further indicated that the mechanical properties of modified bamboo increased by approximately 50%, primarily due to the denser cell wall structure achieved under neutral reaction conditions. In summary, in situ polymerization of NMA presents a promising modification method with broad application potential [53]. Dong et al. used citric acid (CA) and 1,2,3,4-butanetetracarboxylic acid (BTCA) to chemically modify bamboo strips. The results showed that the esterification reaction between CA/BTCA and bamboo components not only improved the mildew resistance and termite resistance of bamboo but also improved the dimensional stability and thermal stability of bamboo [54].

Ma et al. proposed an anticorrosive method of coating an epoxy resin on the surface of bamboo sheets to prevent the corrosion of bamboo sheets and carried out four surface modifications: heat treatment, alkali treatment, coupling treatment, and acetylation treatment. The results showed that the four modifications had different effects on the surface and material of bamboo, which could reduce the ultimate tension of bamboo and improve the interface bonding between bamboo surface and epoxy resin. They concluded that acetylation was the most effective modification [55]. Hung et al. prepared different degrees of acetylated and butylated bamboo particle/plastic composites (BPPC) by the

plate pressing method and investigated the effects of esterification on the mechanical properties and interfacial properties of BPPC. The results showed that although the weight of acetylated bamboo particles increased by only 2%, the internal bond of BPPC and the wood screw-holding strength were significantly increased after acetylation, indicating that the interface interaction between bamboo particles and polymer matrix could be enhanced by acetylation [56].

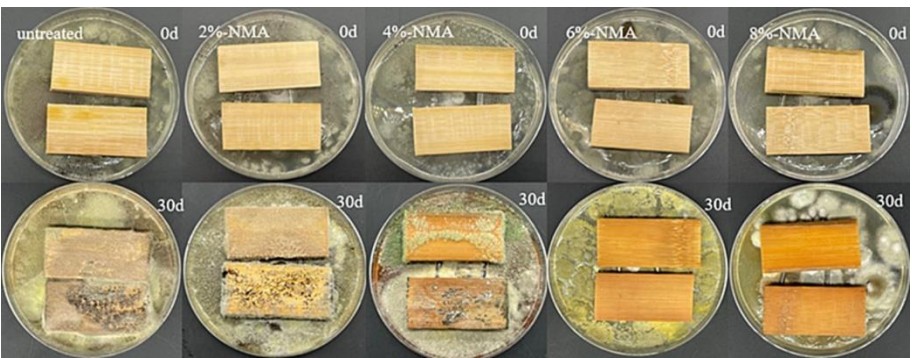

**Figure 3.** The mold resistance of modified and control bamboo [53].

## 4. Impregnation Modification

Impregnation modification involves soaking bamboo in specific chemical liquids to allow it to absorb the modifiers. This process can enhance bamboo resistance to corrosion, cracks, and improve its overall appearance. Impregnation modification can also be employed to confer bamboo with fire resistance, insect repellent properties, energy storage, and other special characteristics, making it suitable for various specialized applications [57–60].

For example, Rao et al. employed three techniques to address the cracking issue of bamboo: impregnation with a polyethylene glycol (PEG)-1000 solution, heat treatment using paraffin alone, and a combination of PEG impregnation and paraffin heat treatment (PEG-PH). The treated bamboo specimens were subjected to a 26-week exposure period to assess crack development. Results revealed that cracks began appearing in the PEG-PH treated bamboo after 22 weeks, whereas the control group exhibited cracking after only 2 weeks. As for the mechanism, PEG can diffuse into the fresh bamboo through the concentration difference instead of water, providing the bamboo with dimensional stability, while the paraffin can form a protective layer on the surface of the bamboo, protecting it from water and moisture. Hence, combining PEG impregnation with paraffin heating proves to be an effective and viable method to prevent bamboo cracking [61]. Su et al. prepared a hydrophobic rosin ethanol solution and impregnated bamboo with vacuum pressure at room temperature to improve the limitations caused by bamboo hydrophilicity. After rosin treatment, the water absorption of bamboo decreased by 24.7%, and the radial and tangential expansion coefficients of bamboo from air drying to water saturation decreased by 23.11% and 21.36%, respectively. After rosin treatment, the surface wettability of bamboo is reduced, and the contact angle is 93°, indicating that hydrophobicity is greatly increased, which is beneficial to its application in building materials [62]. The mechanism and effect of rosin impregnation modification shown in Figure 4. It can be seen that the three infrared characteristic peaks (2931, 2868, and 1693 cm$^{-1}$) of bamboo were enhanced after rosin treatment, indicating that rosin was successfully impregnated into bamboo. In addition, rosin forms a continuous uniform film on the surface of bamboo, covering the main water channels in the bamboo body, and rosin itself is hydrophobic, resulting in significantly improved hydrophobicity of the modified bamboo. Yu et al. impregnated silver-loaded thermal-nanogels, an antifungal agent, into bamboo strips using two methods of air compression and vacuum compression. The optimum parameters of impregnation treatment were impregnation time 90 min, impregnation concentration 0.90 wt%, and pressure 0.5 MPa. The results of the mildew control test showed that the modified bamboo strips had a good mildew control effect and could be used in more fields [63].

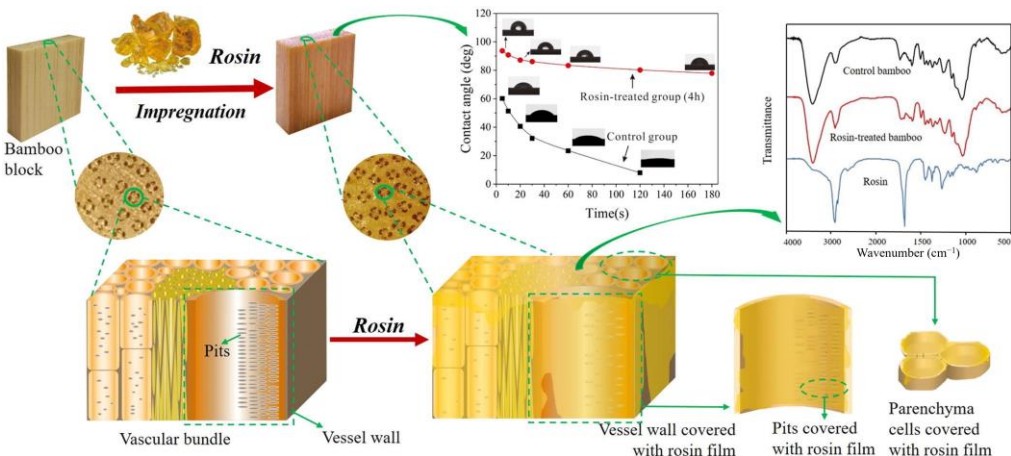

**Figure 4.** Mechanism and effects of rosin impregnation modification [62].

## 5. Plasma Modification

Plasma modification is an advanced technique used to enhance the surface characteristics of bamboo through plasma treatment. This process can improve its wettability, enhance adhesion, and refine dyeing capabilities, among other benefits. Plasma modification allows for the customization of bamboo's surface properties, making it suitable for more sophisticated applications [64,65].

For example, oxygen plasma technology was employed by Rao et al. to functionalize carbonized and non-carbonized bamboo surfaces for the production of advanced bamboo composite materials. Oxygen plasma treatment effectively induced surface oxidation and the formation of oxygen-containing groups, leading to significant alterations in the surface structure of bamboo. This treatment greatly enhanced the surface wettability and interface bonding strength of bamboo, resulting in remarkable improvements in its physical and mechanical properties as a composite material. The SEM images of the untreated and treated bamboo are shown in Figure 5. Specifically, the modulus of MOR reached 170 MPa after treatment, increasing by 47% compared to untreated bamboo composites [66]. However, plasma treatment has timeliness; that is, the bamboo surface will gradually return to the original state after treatment, so it is necessary to minimize the delay between plasma treatment and further processing. Peng et al. applied silver nanoparticles onto bamboo pulp fabric that had undergone plasma pretreatment and conducted a series of characterizations. The results revealed that, when compared to silver nanoparticles applied to bamboo pulp fabric without plasma pretreatment, the plasma-treated bamboo pulp fabric exhibited a higher deposition of silver nanoparticles. These nanoparticles were evenly distributed across the bamboo pulp fabric. Furthermore, the bamboo pulp fabric coated with silver nanoparticles following plasma pretreatment displayed excellent UV protection capabilities, strong resistance to laundering, and hydrophobic properties [67]. The laminated bamboo lumber (LBL) with superior performance through different forming methods were selected by Wu et al., followed by $O_2$ plasma modification. With $O_2$ plasma modification times of 6 min and 12 min, the wet bonding strength of LBL products increased by 58.58% and an impressive 75.69% when compared to conventional plywood. Additionally, the wettability of both green bamboo and yellow bamboo significantly improved following $O_2$ plasma modification [65].

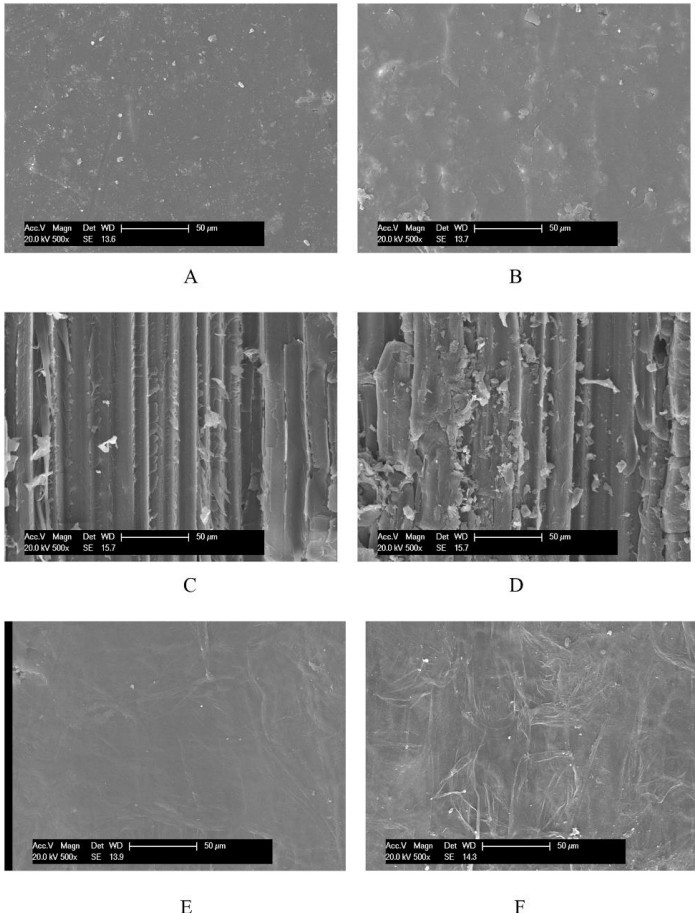

**Figure 5.** SEM images of bamboo surface structure: (**A**) untreated, and (**B**) plasma treated bamboo skin, (**C**) untreated and (**D**) plasma treated bamboo pulp, (**E**) untreated and (**F**) plasma treated bamboo pith [66].

## 6. Other Modifications

In addition to the modification methods mentioned above, emerging techniques such as enzymatic modification, microwave treatment, and nanotechnology treatment are gaining prominence, injecting new vitality into the field of bamboo modification [68–70]. These novel modification methods offer increased potential for enhancing bamboo performance and adaptability. Depending on specific requirements and application areas, different modification techniques can be chosen. This field is continuously evolving, and we can expect more innovative methods to emerge in the future, further expanding the possibilities for the widespread utilization of bamboo across various domains.

### 6.1. Enzymatic Modification

Enzymatic and fungal treatments employ biological approaches to process bamboo by utilizing specific enzymes or fungi, which break down, alter, or degrade the bamboo's cellulose and lignin, thereby enhancing its workability, strength, and durability.

For example, Zhao et al. undertook a meticulous screening of fungal strains, culminating in the identification of Paracremonium sp. LCB1, Clonostachys compactiuscula LCD1, and C. compactiuscula LCN1, which demonstrated the remarkable capability of producing a plethora of ligninolytic enzymes. These strains, when co-cultivated, exhibited a remarkable capacity to degrade hemicellulose and lignin within a relatively concise fermentation period of merely 3–5 days, while the activity of cellulase remained conspicuously subdued. Experimental findings unequivocally demonstrated that under meticulously controlled conditions encompassing temperature (30 °C), pH (5), incubation duration (40 days), and a

precise liquid-to-solid ratio of 1:25.5, the joint cultivation of LCB1 and LCN1 as a bamboo pretreatment method resulted in a striking 76.37% reduction in lignin content. Furthermore, the lignin-to-cellulose loss ratio soared above the 10-fold mark, unequivocally establishing that the co-cultivation of LCB1 and LCN1 represents a highly effective strategy for lignin extraction from bamboo materials [71]. The process of lignin extraction improves fiber quality, reduces lignin content, and enhances the suitability of bamboo for various applications, such as papermaking, textiles, and composite materials. Additionally, the extracted lignin finds broad applications in other fields, such as carbon fibers and antioxidants [72]. Chen et al. utilized a natural organic enzyme-7F (NOE-7F) biological solution for the pretreatment of bamboo. The findings demonstrated that NOE-7F effectively eliminated hemicellulose from bamboo, while also influencing the crystalline structure of cellulose to enhance biomass uniformity. Consequently, the pre-treated bamboo exhibited potential for bioethanol production through enzymatic hydrolysis. Moreover, compared to raw bamboo, the ignition and burnout temperatures of pre-treated bamboo were elevated, indicating reduced reactivity and improved storage safety [73]. In addition, El-Khatib et al. used brewer's yeast suspension to treat bamboo and bamboo-based knitted fabrics. After the optimization of the treatment process, the wettability of bamboo-based knitted fabrics was greatly improved, which was conducive to further dyeing [74]. Liu et al. treated natural bamboo fibers with four enzymes, pectin lyase, xylanase, laccase, and cellulase. The experimental results showed that the content of non-cellulose substances in the modified fibers was significantly reduced, and the fineness was also significantly improved [75]. All these results show that enzymes play an indispensable role in the modification of bamboo-based materials, making bamboo-based materials have more extensive application value.

### 6.2. Microwave Modification

Microwave modification, on the other hand, rapidly heats the interior of bamboo to promote chemical reactions and structural changes, leading to improvements in its physical properties, water resistance, and durability.

Lv et al. delved into the intricate changes brought about by microwave irradiation on the cellular structure of bamboo materials. This treatment yielded a sleeker surface on the samples, instigating the deformation of thin-walled cells while concurrently augmenting porosity. Microwave irradiation effectively reoriented microfibrils through the formation of hydrogen bonds, concurrently resulting in a reduction of the O-H stretching peak at 3440 cm$^{-1}$. Furthermore, this treatment engendered the formation of both intermolecular and intramolecular bonds, ultimately enhancing the uniformity of microfibrils. This intricate process was accompanied by the condensation reaction of surface hydroxyl groups and the formation of intramolecular hydroxyl groups within the amorphous region, thereby contributing to a more orderly alignment of the fibers [76]. Improved fiber alignment enhances the mechanical performance of bamboo products, making them stronger and more suitable for structural applications. In addition, they also studied the shrinkage rate and chemical properties of bamboo stems after microwave drying [77]. Bamboo was selected as a one-meter-long sample, with diameters ranging from 30 to 40 mm, wall thickness ranging from 3.8 to 4.3 mm, and the water content was about 65%. The samples were treated with microwave equipment at 80 °C until the moisture content reaches 10%. The results showed that the shrinkage rate of microwave-dried bamboo was 3.88%, which was lower than that of natural bamboo (4.36%). Liu et al. first delignified natural bamboo, then impregnated it in resin, and finally shrink it after microwave heating to form a dense and uniform bamboo-based composite material. Compared with natural bamboo, the stability of the composite material is better, and the flexural strength and compressive strength of the bamboo composite material are increased by 80.9% and 42.9%, respectively, making it a prospective material for new furniture or construction [78]. Rai et al. used partially delignified and microwave-treated bamboo to prepare mechanically robust, water-stable, and biodegradable straws. The microwave-treated bamboo-based straws possess water stability of up to 16 h and a contact angle of 87.8°, indicating low wettability and long-term

water stability. After treatment, the tensile strength, Young's modulus and bending strength were increased by 59.3 MPa, 988 MPa, and 13.9 MPa, respectively. Compared to plastic straws, microwave-irradiated bamboo straws significantly slow down the trend of global warming during the production process. Therefore, microwave treatment is a fast and low-cost physicochemical modification strategy for bamboo-based materials [79].

*6.3. Nanomaterial Modification*

Nanomaterial modification involves introducing nanoparticles or nanostructures to enhance bamboo's capabilities. For example, Sun et al. synthesized $TiO_2$ sol via a sol-gel method under low-temperature conditions and prepared $TiO_2$ crystalline films with diameters ranging from about 40 to 90 nm. They investigated the impact of temperature on film crystalline morphology, antimicrobial performance, and mold resistance when coating bamboo. The results indicate that modified bamboo primarily underwent rutile-phase nano-$TiO_2$ film coverage at low-temperature solutions (20, 60, and 105 °C). The nanoscale $TiO_2$-modified bamboo retained the natural wood's color, texture, and structure while significantly enhancing its antimicrobial properties from non-antimicrobial to effectively combating E. coli, achieving over a 99% sterilization rate. Furthermore, its resistance to mold increased more than tenfold. Therefore, this method offers a promising approach for enhancing bamboo's functionality, with implications for the protection and improvement of other biological materials [80]. Wang et al. soaked the dehydrated bamboo scaffold in graphene oxide (GO) suspension and degassed it under vacuum for 10 min. Then, hydrophobic $SiO_2$ coating was sprayed on the surface of the GO/bamboo scaffold, and the sample was hot-pressed at 25 MPa and 120 °C for 8 h. Due to the well-preserved bamboo scaffold and the strong hydrogen bond between the bamboo fiber and the GO nanosheets, the bamboo-composite has extremely high tensile strength (641.6 MPa), excellent flexural strength (428.4 MPa), and excellent toughness (17.5 MJ/m$^3$). It is about 480%, 250%, and 360% higher than natural bamboo, respectively, and is expected to be widely used in the field of practical structural engineering [81]. More recently, Ba et al. transformed natural bamboo into high-performance structural materials through high-pressure densification of in situ hydrothermal synthesis of $TiO_2$ and delignified bamboo. It was found that the introduction of nanoscale $TiO_2$ significantly increased the oxidation degree and hydrogen bond formation of bamboo materials, and the resulting bamboo decorated with $TiO_2$ material had more than twice the flexural strength and elastic stiffness of natural bamboo. This work extends synthetic enhancement strategies for natural materials [82]. Similarly, Tang et al. also demonstrated that nano $TiO_2$ gel coating on the surface of bamboo after heat treatment has improved the performance of bamboo [83].

Overall, these modification methods have unlocked new avenues for the widespread use of bamboo, positioning it as a green and sustainable material with the potential to drive innovation and environmental responsibility across various sectors. This further underscores the versatility of bamboo as a multifunctional material and its critical role in sustainable material development. Bamboo, with its outstanding renewability, and continually evolving modification techniques, not only widens its application scope but also promises boundless potential for the future of green materials. It will continue to play a pivotal role in areas such as construction, furniture, and environmental conservation, contributing to a more sustainable future.

## 7. Conclusions

The pursuit of efficient modification methods to overcome bamboo's inherent limitations has consistently been a goal for researchers. In this paper, we present a comprehensive review of various bamboo modification methods, conducting an in-depth analysis of their strengths and weaknesses, while also elucidating the fundamental mechanisms behind the modifications. Looking towards the future, there is a compelling need to develop even more efficient modification methods, such as high-efficiency enzyme treatment, advanced nanotechnology, and composite modifications, to expand the range of bamboo applications.

This review provides valuable insights and references for future research in the bamboo sector, with the potential to drive the widespread utilization of bamboo. This, in turn, can better cater to the diverse needs of various fields, advancing the realization of sustainability and environmental conservation principles.

**Author Contributions:** Conceptualization, Z.Z. and N.Y.; writing—original draft preparation, Z.Z., N.Y., X.J., X.Z., R.X. and S.C.; writing—review and editing, Z.Z., C.L., L.X. and Z.L.; supervision, project administration, L.X. and Z.L. All authors have read and agreed to the published version of the manuscript.

**Funding:** This research was funded by the National Key R&D Program of China [2023YFE0108300], the Natural Science Foundation of Jiangsu Province (BK20221336), Jiangsu Agricultural Science and Technology Innovation Fund [CX(21)1010 and CX (23)3060], Jiangxi Forestry Bureau Forestry Science and Technology Innovation Special Project [No. 202240], and open funding from the Key Laboratory for Protected Agricultural Engineering in the Middle and Lower Reaches of Yangtze River [ZX(23)3104].

**Conflicts of Interest:** The authors declare no conflict of interest.

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
