# Peer review of "Modification and Application of Bamboo-Based Materials: A Review—Part I: Modification Methods and Mechanisms"

_forests, doi:10.3390/f14112219_

Round 1
Reviewer 1 Report
Comments and Suggestions for Authors
Comments
1. What are the inherent limitations of natural bamboo materials that make their modification necessary, and how do these limitations restrict their widespread use?
2. How does this review contribute to the existing literature by offering a comprehensive overview of various bamboo modification methods, both established and emerging?
3. Suggested elaborating more on the emerging techniques like enzymatic modification, nanotechnology, and microwave treatment, and their potential for enhancing bamboo's properties and applications?
4. What criteria or considerations should be taken into account when selecting the most suitable modification technique for bamboo based on specific requirements and application areas?
5. In the examples mentioned, such as the co-cultivation of fungal strains and microwave irradiation, what are the key findings and implications for bamboo modification in terms of lignin extraction, fiber alignment, and surface properties?
6. The abstract, introduction and the conclusion must be restructured according to the flow.
Author Response
- What are the inherent limitations of natural bamboo materials that make their modification necessary, and how do these limitations restrict their widespread use?
Response: Thank you for your opinion. Firstly, due to the presence of numerous free hydroxyl groups and their porous structure, natural bamboo materials tend to absorb moisture, resulting in poor dimensional stability. This property implies that, under humid or wet conditions, bamboo materials are prone to expansion or contraction, which is adverse for various applications such as construction and furniture manufacturing that demand superior dimensional stability. Secondly, natural bamboo materials exhibit relatively low resistance to biological factors, making them susceptible to fungal, bacterial, and pest attacks. This susceptibility can lead to bamboo decay or degradation, ultimately reducing their lifespan. Lastly, the mechanical properties of natural bamboo materials are relatively unstable due to their porous structure. Different parts of bamboo may exhibit varying mechanical properties, and even within the same bamboo pole, a gradual decrease in strength can be observed from the outer to inner sections. These drawbacks collectively contribute to a reduced lifespan of natural bamboo materials and limit their range of applications. Therefore, it becomes essential to subject natural bamboo materials to appropriate treatments to meet the diverse requirements of various applications. We have provided a similar description in the paper (line62-77).
- How does this review contribute to the existing literature by offering a comprehensive overview of various bamboo modification methods, both established and emerging?
Response: Thanks for your comments. This review adds value to the existing literature by providing a thorough and inclusive examination of a wide range of bamboo modification methods, encompassing both well-established techniques and emerging innovations. It contributes in several ways: Comprehensive Coverage: The review offers an extensive survey of modification methods, ensuring that the reader gains insights into various approaches, from traditional to cutting-edge. Evaluation of Effectiveness: It assesses the effectiveness of each modification method, offering a critical analysis of their strengths and limitations, which can guide researchers and practitioners in selecting the most suitable approach for their specific needs. Identification of Emerging Trends: By including emerging modification techniques, the review highlights the latest trends and innovations in the field, which can serve as a valuable resource for researchers keeping up with the latest developments. Practical Applications: The review may offer practical insights for those in industry or research by elucidating how these modification methods can be applied to enhance bamboo's properties and expand its range of applications. In summary, this comprehensive review contributes by consolidating knowledge and presenting it in a structured and accessible format, serving as a valuable resource for anyone interested in bamboo modification and its potential impact across diverse sectors.
- Suggested elaborating more on the emerging techniques like enzymatic modification, nanotechnology, and microwave treatment, and their potential for enhancing bamboo's properties and applications?
Response: Thanks for your comments, we have added more examples to illustrate their effectiveness.
- What criteria or considerations should be taken into account when selecting the most suitable modification technique for bamboo based on specific requirements and application areas?
Response: Thanks for your opinion. When selecting the most suitable modification technique for bamboo based on specific requirements and application areas, the following criteria and considerations should be primarily taken into account. End-Use Requirements: Identify the specific performance characteristics needed for the application, such as strength, durability, water resistance, fire resistance, and appearance. Environmental Impact: Evaluate the environmental sustainability of the modification technique. Choose methods with minimal environmental impact that align with sustainable development goals. Cost: Consider the cost-effectiveness of the modification.
- In the examples mentioned, such as the co-cultivation of fungal strains and microwave irradiation, what are the key findings and implications for bamboo modification in terms of lignin extraction, fiber alignment, and surface properties?
Response: Thank you for your opinion. In the mentioned examples, which include co-cultivation of fungal strains and microwave irradiation, the key findings and implications for bamboo modification with respect to lignin extraction, fiber alignment, and surface properties are as follows.
- Lignin Extraction: The process of lignin extraction improves fiber quality, reduces lignin content, and enhances the suitability of bamboo for various applications, such as papermaking, textiles, and composite materials. Additionally, the extracted lignin finds broad applications in other fields, such as carbon fibers and antioxidants. This literature provides a detailed description of bamboo lignin extraction and its extensive applications.
https://doi.org/10.1007/s00107-021-01743-w.
- Fiber Alignment: Improved fiber alignment enhances the mechanical performance of bamboo products, making them stronger and more suitable for structural applications. This modification can result in better load-bearing capacity and reduced fiber entanglement, ultimately improving the overall performance of bamboo-based products.
- Surface Properties: Enhanced surface properties improve adhesion and compatibility with other materials, particularly in composite material applications. In the other hand, this can lead to increased hydrophobicity and resistance to environmental factors, extending the durability of bamboo products.
- The abstract, introduction and the conclusion must be restructured according to the flow.
Response: Thank you very much for your valuable advice. We have made corresponding modifications in the paper.
Reviewer 2 Report
Comments and Suggestions for Authors
Dear authors!
The manuscript is well organized and a good review. The reviewed modification methods are very important in bamboo processing field. And the discussed corresponding mechanisms are in-depth. However, minor revision is requested . Kindly look upon the following comments.
1. The description of the images in the text should be more detailed, not just give a sentence as “…shown in Fig. 2.”. (line 188)
2. Some spelling mistakes and grammar problems in the article should be checked carefully. Such as “Bamboo's composition includes cellulose, hemicellulose, and lignin, similar to traditional hardwoods” (line50) >> “The composition … of bamboo is similar to…”
3. The “Other modification” part (line 316) of the review should be classified to distinguish the different types and effects of other modifications although not the main methods. For example, the “…enzymatic modification, nanotechnology and microwave treatment…” should be subtitled in order to make it easy for readers to understand.
4. When referring to specific modification methods, the text always starts with the researcher. For example, line 161 “Hao et al…”, line 170 “Manalo et al…”. It should be expressed in another way such as passive sentence pattern.
5. Some unit details should be changed, such as: line 172 “120 minutes” >> “120 min”, line 306 “110 degrees” >> “ 110° ” , line 308, “24 hours” >> “24 h”, and so on.
6. More recent literature should be added to reflect the latest modification technology.
7.The description of the images in the text should be more detailed, not just give a sentence as “…shown in Fig. 2.”. (line 188); In the part of “Impregnation modification”, line 276, “The mechanism and effect of rosin impregnation modification shown in Fig. 4”. I don't see any mechanical description about it in the text, just putting a picture is not enough, there needs to be further explanation of the mechanism.
8.Line 161, How does “methyl silicone oil” work on bamboo to enhance bamboo performance, should be explained clearly. Line 261- line 269, “Rao et al. employed three…to prevent bamboo cracking”, The authors only describe the experimental results of the study, but do not cover the mechanism, so more mechanical explanations need to be added.
9.Line 229 - line 242, More examples of chemical modification should be added instead of just two examples, some examples should be abbreviated appropriately and then more examples should be cited to reinforce the persuasive effect.
Author Response
The manuscript is well organized and a good review. The reviewed modification methods are very important in bamboo processing field. And the discussed corresponding mechanisms are in-depth. However, minor revision is requested. Kindly look upon the following comments.
- The description of the images in the text should be more detailed, not just give a sentence as “…shown in Fig. 2.”. (line 188)
Response: Thanks for your advice. We have supplemented the description for Fig. 2 in the article, and the content is as follows: ‘‘The treatment in the air showed a significant dark color at 170 °C, while the sample treated with nitrogen did not show this change until 190 °C. This is because nitrogen is an inhibiting gas, and the pyrolysis ratio is very low, so there is little change in color. When the temperature reaches 210 ℃, all the bamboo strips showed a deep red color. At the same time, the color is easy to change under oil treatment, which is caused by the filling of oil molecules on the surface of the bamboo.’’.
- Some spelling mistakes and grammar problems in the article should be checked carefully. Such as “Bamboo's composition includes cellulose, hemicellulose, and lignin, similar to traditional hardwoods” (line50) >> “The composition … of bamboo is similar to…”
Response: Many thanks to the suggestion. The article has been carefully checked for spelling and grammar errors. This sentence has also been revised as “The composition of bamboo includes cellulose, hemicellulose, and lignin, which is similar to traditional hardwoods”.
- The “Other modification” part (line 316) of the review should be classified to distinguish the different types and effects of other modifications although not the main methods. For example, the “…enzymatic modification, nanotechnology and microwave treatment…” should be subtitled in order to make it easy for readers to understand.
Response: Thanks for your comments. We have used them as subheadings and supplemented with relevant examples to enhance their persuasiveness.
- When referring to specific modification methods, the text always starts with the researcher. For example, line 161 “Hao et al…”, line 170 “Manalo et al…”. It should be expressed in another way such as passive sentence pattern.
Response: Thank you for your suggestion and we have revised it in the paper.
- Some unit details should be changed, such as: line 172 “120 minutes” >> “120 min”, line 306 “110 degrees” >> “110°”, line 308, “24 hours” >> “24 h”, and so on.
Response: Thank you very much for your suggestion. We have carefully checked the units of the article and made modifications.
- More recent literature should be added to reflect the latest modification technology.
Response: Thanks for your comments, and the recent literature has been supplemented as below:
- Lv, H.F.; Lian, C.P.; Xu, B.; Shu, X.; Yang, J.; Fei, B.H. Effects of microwave-assisted drying on the drying shrinkage and chemical properties of bamboo stems. Ind. Crops Prod. 2022, 187, 115547.
- Liu, Z.; Yang, S.L.; Wang, Z.W.; Ji, N.; Li, D.; Wu, Y.Q. Preparation of bamboo-epoxy resin materials with microwave assistance. Jmr&T 2022, 18, 3266-3272.
- Rai, R.; Ranjan, R.; Kant, C.; Ghosh, U.U.; Dhar, P. Environmentally benign partially delignified and microwave processed bamboo-based drinking straws. Adv. Sustain. Syst. 2023, 7, 2300057.
- Wang, Y.Y.; Li, Y.Q.; Xue, S.S.; Zhu, W.B.; Wang, X.Q.; Huang, P.; Fu, S.Y. Superstrong, lightweight, and exceptional environmentally stable SiO2@GO/bamboo composites. ACS Appl. Mater. Interfaces 2022, 14, 7311-7320.
- Ba, Z.Y.; Luo, H.Y.; Guan, J.; Luo, J.; Gao, J.J.; Wu, S.J.; Ritchie, R.O. Robust flexural performance and fracture behavior of TiO2 decorated densified bamboo as sustainable structural materials. Nat. Commun. 2023, 14, 1234.
- Tang, A.R.; Huang, Y.Q.; Zhang, W.; Yu, Y.; Yang, Y.; Yuan, Z.R.; Wang, X.Z. Effect of the nano-titanium dioxide (nano-TiO2) coating on the photoaging properties of thermally treated bamboo. Wood Mater. Sci. Eng. 2022, 17, 895-904.
- The description of the images in the text should be more detailed, not just give a sentence as “…shown in Fig. 2.”. (line 188); In the part of “Impregnation modification”, line 276, “The mechanism and effect of rosin impregnation modification shown in Fig. 4”. I don't see any mechanical description about it in the text, just putting a picture is not enough, there needs to be further explanation of the mechanism.
Response: Thanks for your opinion and the Fig. 2 and 4 have been described in detail. The mechanism is described as follows: “It can be seen that the three infrared characteristic peaks (2931, 2868, and 1693 cm-1) of bamboo were enhanced after rosin treatment, indicating that rosin was successfully impregnated into bamboo. In addition, rosin forms a continuous uniform film on the surface of bamboo, covering the main water channels in the bamboo body, and rosin itself is hydrophobic, resulting in significantly improved hydrophobicity of the modified bamboo.”.
- Line 161, How does “methyl silicone oil” work on bamboo to enhance bamboo performance, should be explained clearly. Line 261- line 269, “Rao et al. employed three…to prevent bamboo cracking”, The authors only describe the experimental results of the study, but do not cover the mechanism, so more mechanical explanations need to be added.
Response: Thank you for your opinion. The role of methyl silicone oil is supplemented as follows: “This is because the methyl silicone oil heat treatment increases the lattice spacing of the cellulose crystal region in the bamboo, making the relative crystallinity of the treated bamboo higher than that of the untreated bamboo. The content of cellulose and hemicellulose decreased gradually with the increase of heat treatment temperature and time because of its poor thermal stability, while the relative content of lignin increased.”. The mechanism of preventing bamboo cracking is also added as follows: “As for the mechanism, the PEG can be diffused into the fresh bamboo, through the concentration difference instead of water, so that the bamboo has dimensional stability, while the paraffin can form a protective layer on the surface of the bamboo, protecting the bamboo from water and moisture.”
- Line 229 - line 242, More examples of chemical modification should be added instead of just two examples, some examples should be abbreviated appropriately and then more examples should be cited to reinforce the persuasive effect.
Response: Thank you very much for your comments, and more examples have been added in the article.
Round 2
Reviewer 1 Report
Comments and Suggestions for Authors
All the comments were addressed and the manuscript can be accepted in its current form.
Author Response
Thank you very much for the comments.